# Synthesis and Characterization of High-Entropy-Alloy-Type Layered Telluride *M*Bi_2_Te_4_ (*M* = Ag, In, Sn, Pb, Bi)

**DOI:** 10.3390/ma15072614

**Published:** 2022-04-01

**Authors:** Yuki Nakahira, Seiya Shimono, Yosuke Goto, Akira Miura, Chikako Moriyoshi, Yoshikazu Mizuguchi

**Affiliations:** 1Department of Physics, Tokyo Metropolitan University, 1-1 Minami-Osawa, Hachioji 192-0397, Japan; yuki-nakahira@tmu.ac.jp (Y.N.); y_goto@tmu.ac.jp (Y.G.); 2Department of Materials Science and Engineering, National Defense Academy, Kanagawa 239-8686, Japan; sshimono@nda.ac.jp; 3Faculty of Engineering, Hokkaido University, Kita 13, Nishi 8, Sapporo 060-8628, Japan; amiura@eng.hokudai.ac.jp; 4Graduate School of Advanced Science and Engineering, Hiroshima University, 1-3-1 Kagamiyama, Higashihiroshima 739-8526, Japan; moriyosi@hiroshima-u.ac.jp

**Keywords:** high-entropy alloy, layered compound, synchrotron XRD, PbBi_2_Te_4_

## Abstract

Recently, high-entropy alloys (HEAs) and HEA-type compounds have been extensively studied in the fields of material science and engineering. In this article, we report on the synthesis of a layered system *M*Bi_2_Te_4_ where the *M* site possesses low-, middle-, and high-entropy states. The samples with *M* = Pb, Ag_1/3_Pb_1/3_Bi_1/3_, and Ag_1/5_In_1/5_Sn_1/5_Pb_1/5_Bi_1/5_ were newly synthesized and the crystal structure was examined by synchrotron X-ray diffraction and Rietveld refinement. We found that the *M*-Te2 distance was systematically compressed with decreasing lattice constants, where the configurational entropy of mixing at the *M* site is also systematically increased. The details of structural refinements and the electrical transport property are presented.

## 1. Introduction

High-entropy alloys (HEAs) are alloys containing five or more elements with a concentration range of 5–35 at% [1,2]. Due to the effects of high configurational entropy of mixing (Δ*S*_mix_), which is defined as Δ*S*_mix_ = −*R* Σ*_i_ c_i_* ln *c_i_*, where *c_i_* and *R* are the compositional ratio and the gas constant, respectively, the HEAs have exhibited high performance, such as high stability or toughness under extreme conditions [1]. Since 2018, we have developed HEA-type compounds with a complicated structure (more complicated alloy-based HEAs) [3]. In such compounds with two or more crystallographic sites, random and scattered atomic bond lengths are expected due to the introduction of a HEA-type site, which is a crystallographic site satisfying the definition of HEA by alloying. In the BiS_2_-based *RE*(O,F)BiS_2_ (*RE*: rare earth) and cuprate *RE*123 (*RE*Ba_2_Cu_3_O_7-*d*_) layered superconductors [4,5,6], in which the *RE* site is alloyed with five elements, we found that the introduction of HEA site does not largely affect electronic states because the superconducting transition temperature (*T*_c_) of HEA-type samples was comparable to that for zero- or low-entropy samples. However, in BiS_2_-based *RE*O_0.5_F_0.5_BiS_2_, the modification of local structure, the decrease in in-plane atomic displacement parameter *U*_11_, was observed, and the superconducting properties were improved by the increase in Δ*S*_mix_ at the *RE* site [7]. This trend suggests that the presence of the HEA site affects the local structure of the layered system. For the cases of low-dimensional structure, we studied quasi-two-dimensional *Tr*Zr_2_ (*Tr*: transition metal) and found that the *T*_c_ is insensitive to Δ*S*_mix_ [8,9]. However, the entropy dependent evaluation of the sharpness of the specific heat jump at *T*_c_ in *Tr*Zr_2_ samples clearly showed anomalous broadening as Δ*S*_mix_ increased [10]. The results suggest that the HEA states would affect the superconducting gap opening. More recently, HEA-type van-der-Waals layered superconductors were designed and synthesized [11]. For the cases of three-dimensional structures, we have investigated the effect of the introduction of HEA sites in NaCl-type *M*Te [12,13,14] and A15-type Nb_3_*X* and V_3_*X* [15,16]. In both systems, the increase in Δ*S*_mix_ resulted in the decrease in *T*_c_. However, anomalous robustness of superconductivity under high pressure was found to be induced by increasing Δ*S*_mix_ in *M*Te with a CsCl-type structure (high-pressure phase) [17], which is similar to that observed in a HEA superconductor Ti-Zr-Hf-Nb-Ta [18]. Therefore, three-dimensional HEA-type compounds will be important to find out the universality in HEA-type superconducting materials. In addition, in V_3_*X*, intrinsic phase separation, which results in the satisfaction with the condition of compositionally-complexed-alloy (CCA) states, was observed, and the upper critical field was improved by the formation of the CCA states [17]. Therefore, the three-dimensional system is also unique in the effects of HEA in compounds. Furthermore, recent studies on HEA effects in thermoelectric materials have suggested that the HEA effects can be very useful for achieving high thermoelectric performance in three-dimensional and quasi-two-dimensional systems [19,20,21,22,23]. Based on those interests in HEA-type compounds with functionality, we decided to study the effect of HEA in a layered system PbBi_2_Te_4_. Recently, pressure-induced superconductivity was reported in Ref. [24]. Having considered the modification of local structure in HEA-type compounds and the positive effects on superconductivity in the layered *RE*O_0.5_F_0.5_BiS_2_ system [7], we expected the emergence of superconductivity by the HEA effects in *M*Bi_2_Te_4_ where *M* is occupied by Ag, In, Sn, Pb, and Bi. Unfortunately, we could not observe superconductivity in the HEA-type (Ag_1/5_In_1/5_Sn_1/5_Pb_1/5_Bi_1/5_)Bi_2_Te_4_, but the observed structural changes would be useful for material design with the concept of HEA.

## 2. Materials and Methods

The samples of PbBi_2_Te_4_ (nominal Δ*S*_mix_ = 0), (Ag_1/3_Pb_1/3_Bi_1/3_)Bi_2_Te_4_ (MEA, nominal Δ*S*_mix_ = 1.10 *R*), and (Ag_1/5_In_1/5_Sn_1/5_Pb_1/5_Bi_1/5_)Bi_2_Te_4_ (HEA: nominal Δ*S*_mix_ = 1.61 R) were synthesized using solid-state reaction. Since the optimal annealing condition changed according to the composition (and possibly due to the difference in entropy of mixing), we optimized the annealing conditions for the MEA and HEA samples. For example, using the same condition as MEA for the HEA composition resulted in the scattering of Ag concentration in the obtained sample. To avoid inhomogeneous compositions, we optimized the condition, and a detailed investigation was performed on the best samples. The raw materials were wires of Ag (99.9%, Kojundo Chemical Laboratory, Sakado, Japan) and powders of In (99.99%, Kojundo Chemical Laboratory, Sakado, Japan), Sn (99.99%, Kojundo Chemical Laboratory, Sakado, Japan), Pb (99.9%, Kojundo Chemical Laboratory, Sakado, Japan), Bi (99.999%, Kojundo Chemical Laboratory, Sakado, Japan), and Te (99.999%, Kojundo Chemical Laboratory, Sakado, Japan). For PbBi_2_Te_4_, the stoichiometric mixture of raw materials was sealed in an evacuated quartz tube (with a pressure lower than 0.5 Pa), heated at 1093 K for 24 h in a box-type electric furnace and cooled to 373 K with a cooling rate of −5 K/h. Finally, the sample was furnace-cooled after all heat treatments. A similar procedure, except for annealing conditions, was applied to other samples and the heating condition was optimized by checking X-ray diffraction (XRD) patterns collected on Miniflex600 (RIGAKU, Akishima, Japan; CuK_α_). For the MEA sample, the stoichiometric mixture of raw materials was heated at 1273 K for 24 h and cooled to 673 K with a cooling rate of −12 K/h followed by holding the temperature at 673 K for 50 h in an evacuated quartz tube. For the HEA sample, the stoichiometric mixture of raw materials was heated at 1323 K for 10 h and cooled to 623 K with a cooling rate of −12 K/h, followed by holding the temperature at 623 K for 50 h in an evacuated quartz tube. The obtained samples contained single crystals with a shiny plane. The chemical compositions of samples were evaluated by energy-dispersive X-ray spectrometry (EDX) using an SEM-EDX system (TM3030 scanning tunneling microscope (Hitachi Hightech, Tokyo, Japan) equipped with an EDX spectrometer (SwiftED, Oxford, UK) on a shiny plane of the samples. The composition was evaluated by taking an average of 13 points. The synchrotron X-ray diffraction (SXRD) patterns were obtained using the multiple microstrip detector MYTHEN system [25] at the BL02B2 beamline of the synchrotron facility SPring-8 (Harima, Japan) with X-ray of 25 keV (wavelength, λ = 0.496118(1) Å). We used powder samples for SXRD, and the powders were sealed in a boro-silicate glass capillary with a diameter of 0.1 mm. The Rietveld structure refinements were performed using the *JANA2020* software [26]. The image of the crystal structure was drawn using the *VESTA* software [27]. The electrical resistivity was measured by the four-probe method on a Physical Property Measurement System (PPMS, Quantum Design, San Diego, CA, USA). For the PbBi_2_Te_4_ and HEA samples, the resistivity measurements were performed with selected crystals having a flat plane. The size of the PbBi_2_Te_4_ and HEA samples is 0.1 × 1.2 × 1.7 mm^3^ and 0.4 × 1.2 × 2.6 mm^3^, respectively. The terminals were fabricated using Ag paste and Au wires with a diameter of 25 μm. Magnetization was measured by a superconducting quantum interference device (SQUID) magnetometer on MPMS3 (Quantum Design, San Diego, CA, USA).

## 3. Results

The SEM image for the HEA sample is shown in Figure 1d. In Table 1, actual compositions for the *M* site estimated by EDX are listed. In the EDX analysis, the Bi site was assumed to be fully occupied with Bi, and the *M*-site composition was analyzed. The obtained values for the MEA and HEA samples are Ag_0.18_Pb_0.32_Bi_0.50_Bi_2_Te_4_ and Ag_0.14_In_0.15_Sn_0.25_Pb_0.14_Bi_0.32_Bi_2_Te_4_, respectively. We found that even for PbBi_2_Te_4_, Bi-rich composition with Δ*S*_mix_ = 0.61*R* for the *M* site was obtained. A similar trend of Bi-rich composition was observed in Reference [22]. For the MEA and HEA samples, a slight deviation of the actual composition from the nominal ones was observed, but the expected Δ*S*_mix_ was almost preserved: Δ*S*_mix_ = 1.02*R* and 1.55*R* for MEA and HEA, respectively.

The SXRD patterns and the Rietveld fitting results are shown in Figure 1. In the multi-phase Rietveld refinements, the impurity amount was also refined. For PbBi_2_Te_4_ (Figure 1a), two impurity phases of PbBi_4_Te_7_ (21%) and PbTe (13%) were detected. The MEA and HEA samples included 38% and 18% of the PbBi_4_Te_7_ impurity. The formation of impurity phases would be understood by the presence of three phases (PbTe, PbBi_2_Te_4_, and PbBi_4_Te_7_) in the ternary phase diagram at similar temperature regions. Although we tried to optimize annealing conditions, the amount of impurity phases could not be reduced. In the HEA sample, however, the amount of impurity was the lowest, which would suggest that the *M*Bi_2_Te_4_ phase was stabilized by the HEA effect at the *M* site [28].

The structural parameters obtained from the Rietveld refinements are listed in Table 2a–c, and the typical bond distances are plotted in Figure 2 as a function of Δ*S*_mix_/*R* (EDX) at the *M* site. With increasing Δ*S*_mix_, the *M*-Te2 distance decreases, which is probably well explained by the change in average ionic radius. The shrinkage of the *M*-Te2 distance affects other structural parameters; the Te2-*M*-Te2 angle increases, the *M*Te_6_ octahedron is compressed along the *c* axis, and the BiTe_6_ octahedron is compressed along the *ab* plane. Although those changes would be caused by chemical pressure effects, which is the shrinkage of unit cells and/or bonds by the decrease in the average ionic size at the *M* site, we found an interesting trend in the isotropic displacement parameter (*U*_iso_) at the M site. With increasing Δ*S*_mix_, *U*_iso_ at the *M* site clearly decreases (Table 2). In the Discussion, we briefly discuss the origin and commonality of this trend in HEA-type compounds.

In Figure 3, the temperature dependences of electrical resistivity (*ρ*) normalized by that at 300 K (*ρ* (*T*)/300 K) are displayed. For both the PbBi_2_Te_4_ and HEA samples, metallic conductivity, which is regarded by the decrease in *ρ* with decreasing temperature, is observed. For the HEA sample, the temperature dependence is weak and the residual resistivity at the lowest temperature is larger than that for PbBi_2_Te_4_. A similar trend was observed in other HEA-type materials as reviewed in Ref. [3]. Therefore, we consider that the HEA state introduced in the *M* site in *M*Bi_2_Te_4_ affects transport properties in this system as well. Although we measured *ρ* and magnetization down to 1.8 K, no superconducting signal was observed. Since the present samples contain impurity phases as described in the refinement part, high quality samples are desired to further discuss the effect of the HEA states on transport properties.

## 4. Discussion

Here, we briefly discuss the origin of the decrease in *U*_iso_ in MEA (Ag_0.18_Pb_0.32_Bi_0.50_Bi_2_Te_4_) and HEA (Ag_0.14_In_0.15_Sn_0.25_Pb_0.14_Bi_0.32_Bi_2_Te_4_). When one or more sites of compounds were substituted by different elements, an increase in disorder on atomic distance is expected, and off-centering of the atomic position is also expected. Although the increase in disorder due to element substitutions should strongly depend on the type of compounds (crystal structure), an increase in *U*_iso_ is simply expected for compounds with element solution. However, in this system, *U*_iso_ decreases with increasing Δ*S*_mix_ as shown in Figure 4. We consider that the decrease in *U*_iso_ is correlating with the change in lattice vibration. According to Ref. [29], PbBi_2_Te_4_ possesses anharmonic lattice vibration. In our recent study on a BiS_2_-based layered system RE(O,F)BiS_2_, we showed that lattice anharmonicity is enhanced in the low-entropy region, and is suppressed in the middle-to-high entropy region [30]. According to those facts, we consider that the decrease in *U*_iso_ in *M*Bi_2_Te_4_ is also related to the suppression of the anharmonic vibration. Hence, the synthesis and structural characterization for *M*Bi_2_Te_4_ with different Δ*S*_mix_ would give a strategy to tune lattice vibration in functional materials.

## 5. Conclusions

We synthesized a new layered system *M*Bi_2_Te_4_ where low-, middle-, and high-entropy states are present in the *M* site: the samples with nominal *M* = Pb, Ag_1/3_Pb_1/3_Bi_1/3_, and Ag_1/5_In_1/5_Sn_1/5_Pb_1/5_Bi_1/5_ were synthesized by solid-state reaction. The crystal structure was examined by synchrotron X-ray diffraction and Rietveld refinement. We found that the *M*-Te2 distance was systematically compressed with decreasing lattice constants, where Δ*S*_mix_ at the *M* site is also systematically increased. We found that *U*_iso_ at the *M* site decreases with increasing Δ*S*_mix_. The result is similar to that observed in other systems with anharmonic lattice vibration. Therefore, we concluded that the increase in Δ*S*_mix_ results in the suppression of anharmonicity in *M*Bi_2_Te_4_.

## Figures and Tables

**Figure 1 materials-15-02614-f001:**
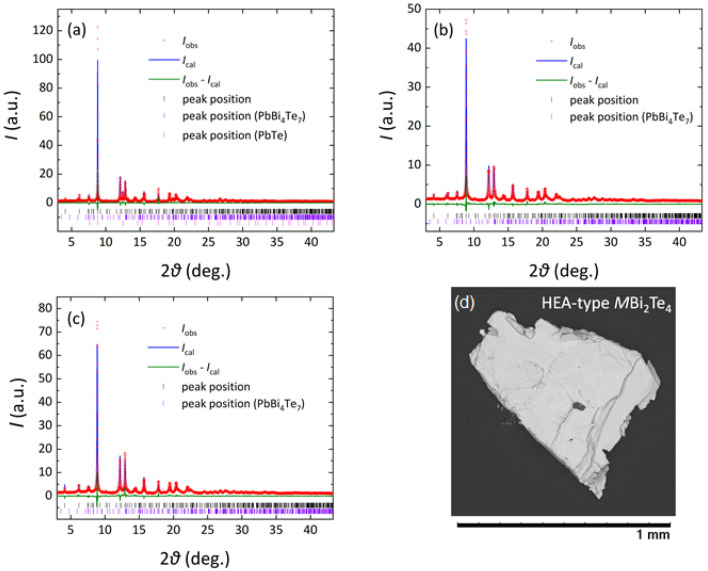
Rietveld profile fitting results for *M*Bi_2_Te_4_: (**a**) PbBi_2_Te_4_ (*d*-spacing range; *d* > 0.67 Å), (**b**) MEA (*d* > 0.66 Å), and (**c**) HEA (*d* > 0.67 Å). The red dots show the measured data (*I*_obs_), and the blue lines are the fitted profiles (*I*_cal_). The green line is the difference curve (*I*_obs_-*I*_cal_), and the black, violet and pink ticks are the peak positions for the main phase (*M*Bi_2_Te_4_) and the impurity of PbBi_4_Te_7_ and PbTe. The PbBi_2_Te_4_ sample included PbBi_4_Te_7_ (21%) and PbTe (13%). The MEA sample included PbBi_4_Te_7_ (38%). The HEA sample included PbBi_4_Te_7_ (16%). (**d**) SEM image of the HEA sample.

**Figure 2 materials-15-02614-f002:**
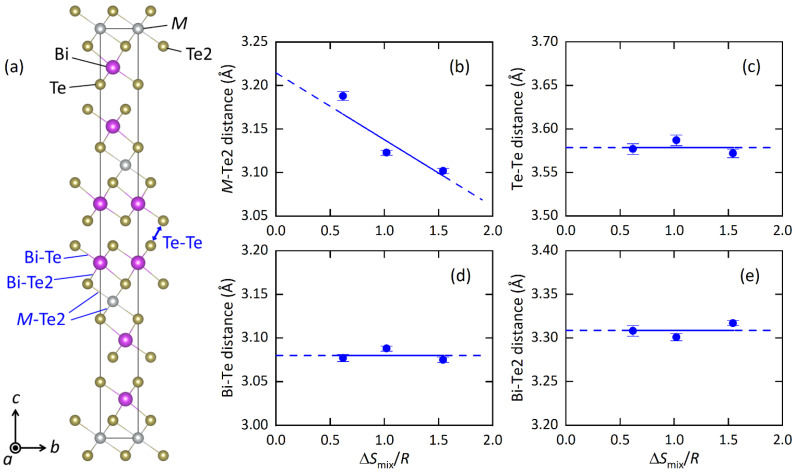
(**a**) Schematic representation of the unit cell of *M*Bi_2_Te_4_, looking along the crystallographic *a*-axis. (**b**–**e**) Atomic distance of *M*-Te2, Te-Te, Bi-Te and Bi-Te2. The blue lines are eye guides.

**Figure 3 materials-15-02614-f003:**
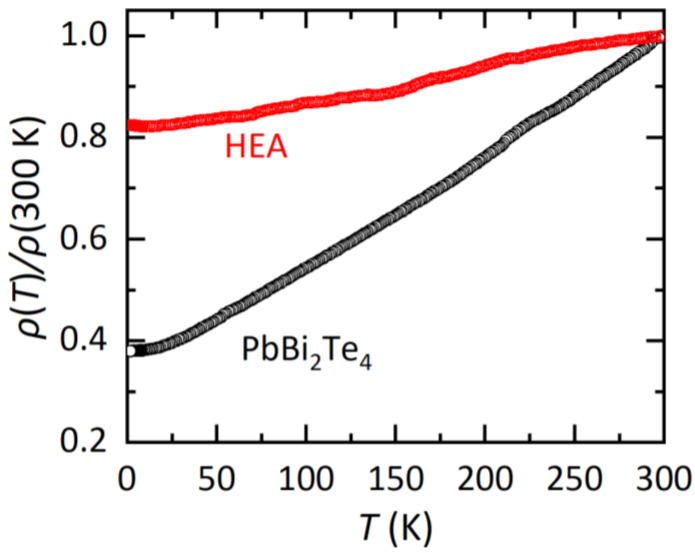
Temperature dependences of electrical resistivity ratio for PbBi_2_Te_4_ and the HEA sample.

**Figure 4 materials-15-02614-f004:**
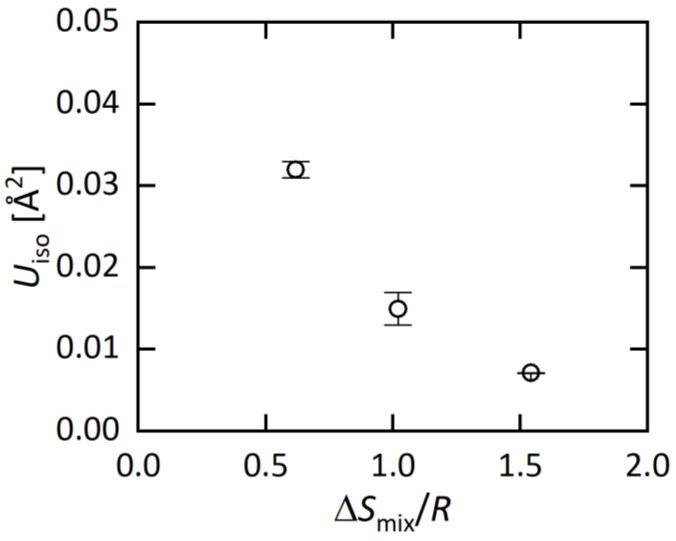
Δ*S*_mix_/*R* dependence of *U*_iso_ at the *M* site.

**Table 1 materials-15-02614-t001:** Occupancies of atoms in the *M* site. The numbers in the brackets are standard deviations.

Nominal Composition	Occupancy of Atom in *M*-Site				
	Ag	In	Sn	Pb	Bi
PbBi_2_Te_4_	–	–	–	0.7(2)	0.3(1)
(Ag_1/3_Pb_1/3_Bi_1/3_)Bi_2_Te_4_	0.18(3)	–	–	0.32(2)	0.50(3)
(Ag_1/5_In_1/5_Sn_1/5_Pb_1/5_Bi_1/5_)Bi_2_Te_4_	0.14(3)	0.15(3)	0.25(3)	0.14(3)	0.32(2)

**Table 2 materials-15-02614-t002:** Refinement results for the SXRD data. In analysis, the occupancy of 3*a* site atoms were fixed nominal values, respectively.

(a) Structure parameters of PbBi_2_Te_4_. Space group *R*$\bar{3}$*m*; *Z* = 3; *a* = 4.43910(8) Å, *c* = 41.6331(12) Å; *wRp* = 0.082; *R*_B_ = 0.116, *R*_F_ = 0.082, goodness-of-fit = 10.67.							
**Atom**	**Site**	**Symmetry**	** *g* **	** *x* **	** *y* **	** *z* **	***U_iso_* (Å^2^)**
Pb	3*a*	$\bar{3}$ *m*	1	0	0	0	0.032(1)
Bi	6*c*	3*m*	1	0	0	0.42913(11)	0.0296(12)
Te	6*c*	3*m*	1	0	0	0.13669(10)	0.0127(7)
Te2	6*c*	3*m*	1	0	0	0.2878(2)	= *U*_iso_ (Te)
(b) Structure parameters of MEA. Space group *R*$\bar{3}$*m*; *Z* = 3; *a* = 4.42165(11) (8) Å, *c* = 41.331(2) Å; *wRp* = 0.054; *R*_B_ = 0.076, *R*_F_ = 0.059, goodness-of-fit = 6.32.							
**Atom**	**Site**	**Symmetry**	** *g* **	** *x* **	** *y* **	** *z* **	***U_iso_* (Å^2^)**
Ag	3*a*	$\bar{3}$ *m*	1/3	0	0	0	0.015(2)
Pb	3*a*	$\bar{3}$ *m*	1/3	0	0	0	= *U*_iso_ (Ag)
Bi	3*a*	$\bar{3}$ *m*	1/3	0	0	0	= *U*_iso_ (Ag)
Bi2	6*c*	3*m*	1	0	0	0.42750(7)	0.0325(8)
Te	6*c*	3*m*	1	0	0	0.13618(9)	0.0100(6)
Te2	6*c*	3*m*	1	0	0	0.28980(12)	= *U*_iso_ (Te)
(c) Structure parameters of HEA. Space group *R*$\bar{3}$*m*; *Z* = 3; *a* = 4.40848(9) Å, *c* = 41.2768(15) Å; *wRp* = 0.063; *R*_B_ = 0.101, *R*_F_ = 0.076, goodness-of-fit = 8.84.							
**Atom**	**Site**	**Symmetry**	** *g* **	** *x* **	** *y* **	** *z* **	***U_iso_* (Å^2^)**
Ag	3*a*	$\bar{3}$ *m*	1/5	0	0	0	0.007110(15)
In	3*a*	$\bar{3}$ *m*	1/5	0	0	0	= *U*_iso_ (Ag)
Sn	3*a*	$\bar{3}$ *m*	1/5	0	0	0	= *U*_iso_ (Ag)
Pb	3*a*	$\bar{3}$ *m*	1/5	0	0	0	= *U*_iso_ (Ag)
Bi	3*a*	$\bar{3}$ *m*	1/5	0	0	0	= *U*_iso_ (Ag)
Bi2	6*c*	3*m*	1	0	0	0.42785(7)	0.0333(8)
Te	6*c*	3*m*	1	0	0	0.13631(8)	0.0143(9)
Te2	6*c*	3*m*	1	0	0	0.29037(9)	= *U*_iso_ (Te)

## Data Availability

The data reported in this article can be provided by corresponding author (Yoshikazu Mizuguchi) through reasonable requests.

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
