# Peer review of "Synthesis and Characterization of High-Entropy-Alloy-Type Layered Telluride MBi2Te4 (M = Ag, In, Sn, Pb, Bi)"

_materials, 2022, doi:10.3390/ma15072614_

Round 1

Reviewer 1 Report

L stands for line.

  1. L28: what do the authors mean by “disordered atomic bonds”?
  2. L29: The authors should define “HEA-type sites”.
  3. L30-32: “we found that the introduction of HEA site does not largely affect electronic states because the superconducting transition temperature (Tc) of HEA-type BiS2-based and RE123 (REBa2Cu3O7-d; RE: rare earth) superconductors is comparable to that for zero- or low-entropy samples.” Unclear description. Is “the introduction of HEA site does not largely affect electronic states” true for all types of layered HEA systems or only for HEA-type BiS2-based and RE123 (REBa2Cu3O7-d; RE: rare earth) superconductors? Please revise the sentence and make it more clear.
  4. L71-78, the authors should provide more details on how the samples were fabricated, e.g., what equipments are used? How the temperature, heating, and cooling rates are controlled? Before heating, how the powders were mixed? In which equipment? The raw material of Ag is in the form of wire. How did the authors obtain powder samples from the wire? The author should describe the synthesis of samples as detailed as possible for others to repeat the experiments. The authors should also state clearly why did they choose such synthesis conditions for each sample.
  5. L78: “The obtained powder samples contained single crystals.” Not sure what do the authors mean. Are the obtained samples single crystal or polycrystal materials?
  6. L78-P80, please give more details of the EDX measurements. How did the authors prepare the samples for EDX? How did the authors perform the EDX measurements? What measuring conditions were used? How did the authors analyze the results of EDX?
  7. L80-83, please give more details of the SXRD measurements. How did the authors prepare the samples for SXRD?
  8. L85-87: the authors should provide more details about the electrical resistivity measurement. “small” is not accurate enough. The authors should state the accurate size of the specimen. How did they prepare the specimen? Are the specimens single crystal or polycrystal?
  9. Results section: for clarity, the authors should also provide the compositions of samples directly obtained by EDX.
  10. Results and discussion section: for clarity, the authors should also mention the name of the samples besides just saying MEA and HEA samples.
  11. Table 1: the authors should state clearly what is the meaning of the numbers in the brackets.
  12. L98-99: the authors should state how did they determine the fractions of impurities.
  13. L104-105, different samples were synthesized under different conditions. Will these different synthesis conditions influence the fraction of impurities?
  14. “Table 1. Occupancies of atoms in the M site. .” There are two periods in the end.
  15. 1: the authors should explain what are Iobs and Ical. It’s difficult to differentiate black and gray ticks, please use different colors.
  16. L114-116: the authors should state how the structural parameters and bond distance were obtained.
  17. L120: for clarity, please give some words on chemical pressure effects.
  18. L120-121: it will be good if the authors provide the ionic size.
  19. L121: please define Uiso.
  20. L122: The authors should mention Fig.3 before Fig.4 in the main text, otherwise changing the orders of the figures.
  21. 2: I suggest the authors move Fig.2(a) to the beginning of the result part.
  22. L139-146: Is the HEA state introduced in the M site the only factor influencing the electrical resistivity? Are there any other factors? Will the impurity content influence the electrical resistivity or not? Will the grain size influence the electrical resistivity or not?
  23. Fig. 3, Why the authors don’t show the curve for the MEA sample? It will be good if the authors also include the results from the MEA sample.
  24. Discussion part: the discussion part is too short. The authors should also discuss the synthesis part, e.g., what are the advantages and disadvantages of the synthesis methods used in this work compared to other methods.

Reviewer 2 Report

Please use the proper words in a few sentences. E.g. (page 98-99) , “For PbBi2Te4 (Fig. 1a), two impurity phases of PbBi4Te7 (21%) and PbTe (13%) were contained.”

From the discussion, the application of the materials synthesized and characterized is not clear. Has the author done any application-oriented measurements? That results should be compared with the results reported in the literature.

In line 91, the author has written, “. In the EDX 91 analysis, the Bi site was assumed to be 2,” As far I know, is it legitimate to do in EDX analysis.

IF possible, please show the microstructure of the surface (SEM) and optical images of the ingots.

Reviewer 3 Report

This article presents interesting data and a concise description of experimental results. I would recommend adding SEM studies of the composite revealing the morphology of the proposed material.

Round 2

Reviewer 1 Report

  1. L29-30: “due to the introduction of a HEA-type site, which is a crystallographic site satisfying the definition of HEA, in compounds with two or more crystallographic sites.” Not sure how a lattice site can satisfy the definition of HEA. They are different types of things, one is lattice sites, one is alloy. The authors should revise the sentence. It’s better that the authors also provide an example in which compound which sites are HEA sites, which are not.
  2. L28-44: the authors are mainly discussed their own work. The authors should discuss others’ work as well.
  3. The authors gave some explanations for the comments in the cover letter but did not include them in the revised manuscript. The author should include these explanations in the revised manuscript as well to provide more information to the readers.

Reviewer 2 Report

For doing EDX, an EDS spectrometer is attached with an SEM instrument. EDX can't be done with an SEM instrument. Please modify your sentence in the manuscript.
